# Perturbative Black Box Variational Inference

**Robert Bamler**[*]
Disney Research
Pittsburgh, USA

**Cheng Zhang**[*]
Disney Research
Pittsburgh, USA

**Manfred Opper**
TU Berlin
Berlin, Germany

**Stephan Mandt**[*]
Disney Research
Pittsburgh, USA

firstname.lastname@{disneyresearch.com, tu-berlin.de}

## Abstract

Black box variational inference (BBVI) with reparameterization gradients triggered the exploration of divergence measures other than the Kullback-Leibler (KL) divergence, such as alpha divergences. In this paper, we view BBVI with generalized divergences as a form of estimating the marginal likelihood via biased importance sampling. The choice of divergence determines a bias-variance trade-off between the tightness of a bound on the marginal likelihood (low bias) and the variance of its gradient estimators. Drawing on variational perturbation theory of statistical physics, we use these insights to construct a family of new variational bounds. Enumerated by an odd integer order $K$, this family captures the standard KL bound for $K = 1$, and converges to the exact marginal likelihood as $K \to \infty$. Compared to alpha-divergences, our reparameterization gradients have a lower variance. We show in experiments on Gaussian Processes and Variational Autoencoders that the new bounds are more mass covering, and that the resulting posterior covariances are closer to the true posterior and lead to higher likelihoods on held-out data.

## 1  Introduction

Variational inference (VI) (Jordan et al., 1999) provides a way to convert Bayesian inference to optimization by minimizing a divergence measure. Recent advances of VI have been devoted to scalability (Hoffman et al., 2013; Ranganath et al., 2014), divergence measures (Minka, 2005; Li and Turner, 2016; Hernandez-Lobato et al., 2016), and structured variational distributions (Hoffman and Blei, 2015; Ranganath et al., 2016).

While traditional stochastic variational inference (SVI) (Hoffman et al., 2013) was limited to conditionally conjugate Bayesian models, black box variational inference (BBVI) (Ranganath et al., 2014) enables SVI on a large class of models. It expresses the gradient as an expectation, and estimates it by Monte-Carlo sampling. A variant of BBVI uses reparameterized gradients and has lower variance (Salimans and Knowles, 2013; Kingma and Welling, 2014; Rezende et al., 2014; Ruiz et al., 2016). BBVI paved the way for approximate inference in complex and deep generative models (Kingma and Welling, 2014; Rezende et al., 2014; Ranganath et al., 2015; Bamler and Mandt, 2017).

Before the advent of BBVI, divergence measures other than the KL divergence had been of limited practical use due to their complexity in both mathematical derivation and computation (Minka, 2005), but have since then been revisited. Alpha-divergences (Hernandez-Lobato et al., 2016; Dieng et al., 2017; Li and Turner, 2016) achieve a better matching of the variational distribution to different regions of the posterior and may be tuned to either fit its dominant mode or to cover its entire support. The problem with reparameterizing the gradient of the alpha-divergence is, however, that the resulting gradient estimates have large variances. It is therefore desirable to find other divergence measures with low-variance reparameterization gradients.

---

[*]Equal contributions. First authorship determined by coin flip among first two authors.

In this paper, we use concepts from perturbation theory of statistical physics to propose a new family of variational bounds on the marginal likelihood with low-variance reparameterization gradients. The lower bounds are enumerated by an order $K$, which takes odd integer values, and are given by

$$\mathcal{L}^{(K)}(\lambda, V_0) = e^{-V_0} \sum_{k=0}^{K} \frac{1}{k!} \underset{\mathbf{z} \sim q}{\mathbb{E}} \Big[ \big( \log p(\mathbf{x}, \mathbf{z}) - \log q(\mathbf{z}; \lambda) + V_0 \big)^k \Big]. \tag{1}$$

Here, $p(\mathbf{x}, \mathbf{z})$ denotes the joint probability density function of the model with observations $\mathbf{x}$ and latent variables $\mathbf{z}$, $q$ is the variational distribution, which depends on variational parameters $\lambda$, and $V_0 \in \mathbb{R}$ is a reference point for the perturbative expansion, see below. In this paper, we motivate and discuss Eq. 1 (Section 3), and we analyze the properties of the proposed bound experimentally (Section 4). Our contributions are as follows.

- We establish a view on black box variational inference with generalized divergences as a form of *biased importance sampling* (Section 3.1). The choice of divergence allows us to trade-off between a low-variance stochastic gradient and loose bound, and a tight variational bound with higher-variance Monte-Carlo gradients. As we explain below, importance sampling and point estimation are at opposite ends of this spectrum.

- We combine these insights with ideas from perturbation theory of statistical physics to motivate the objective function in Eq. 1 (Section 3.2). We show that, for all odd $K$, $\mathcal{L}^{(K)}(\lambda, V_0)$ is a nontrivial lower bound on the marginal likelihood $p(\mathbf{x})$. Thus, we propose the perturbative black box variational inference algorithm (PBBVI), which maximizes $\mathcal{L}^{(K)}(\lambda, V_0)$ over $\lambda$ and $V_0$ with stochastic gradient descent (SGD). For $K = 1$, our algorithm is equivalent to standard BBVI with the KL-divergence (KLVI). On the variance-bias spectrum, KLVI is on the side of large bias and low gradient variance. Increasing $K$ to larger odd integers allows us to gradually trade in some increase in the gradient variance for some reduction of the bias.

- We evaluate our PBBVI algorithm experimentally for the lowest nonstandard order $K = 3$ (Section 4). Compared to KLVI ($K = 1$), our algorithm fits variational distributions that cover more of the mass of the true posterior. Compared to alpha-VI, our experiments confirm that PBBVI uses gradient estimates with lower variance, and converges faster.

## 2 Related work

Our approach is related to BBVI, VI with generalized divergences, and variational perturbation theory. We thus briefly discuss related work in these three directions.

**Black box variational inference (BBVI).** BBVI has already been addressed in the introduction (Salimans and Knowles, 2013; Kingma and Welling, 2014; Rezende et al., 2014; Ranganath et al., 2014; Ruiz et al., 2016); it enables variational inference for many models. Our work builds upon BBVI in that BBVI makes a large class of new divergence measures between the posterior and the approximating distribution tractable. Depending on the divergence measure, BBVI may suffer from high-variance stochastic gradients. This is a practical problem that we aim to improve in this paper.

**Generalized divergences measures.** Our work connects to generalized information-theoretic divergences (Amari, 2012). Minka (2005) introduced a broad class of divergences for variational inference, including alpha-divergences. Most of these divergences have been intractable in large-scale applications until the advent of BBVI. In this context, alpha-divergences were first suggested by Hernandez-Lobato et al. (2016) for local divergence minimization, and later for global minimization by Li and Turner (2016) and Dieng et al. (2017). As we show in this paper, alpha-divergences have the disadvantage of inducing high-variance gradients, since the ratio between posterior and variational distribution enters the bound polynomially instead of logarithmically. In contrast, our approach leads to a more stable inference scheme in high dimensions.

**Variational perturbation theory.** Perturbation theory refers to methods that aim to truncate a typically divergent power series to a convergent series. In machine learning, these approaches have been addressed from an information-theoretic perspective by Tanaka (1999, 2000). Thouless-Anderson-Palmer (TAP) equations (Thouless et al., 1977) are a form of second-order perturbation theory. They were originally developed in statistical physics to include perturbative corrections to the mean-field solution of Ising models. They have been adopted into Bayesian inference in (Plefka, 1982) and were advanced by many authors (Kappen and Wiegerinck, 2001; Paquet et al., 2009; Opper

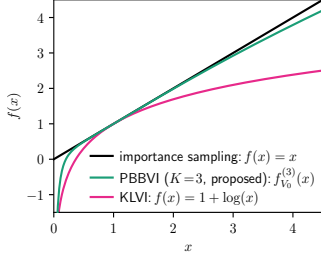

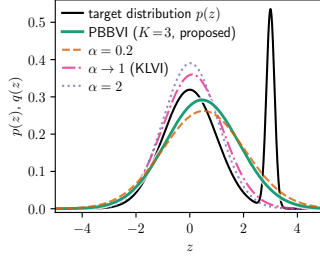

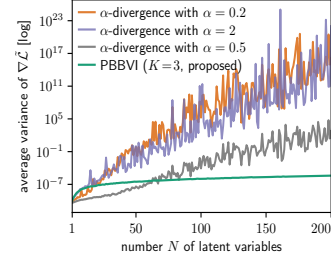

**Figure 1:** Different choices for $f$ in Eq. 4. KLVI corresponds to $f(x) = \log(x) + \text{const.}$ (red), and importance sampling to $f(x) = x$ (black). Our proposed PBBVI bound uses $f_{V_0}^{(K)}$ (green, Eq. 7), which lies between KLVI and importance sampling (we set $K = 3$ and $V_0 = 0$ for PBBVI here).

**Figure 2:** Behavior of different VI methods on fitting a univariate Gaussian to a bimodal target distribution (black). PBBVI (proposed, green) covers more of the mass of the entire distribution than the traditional KLVI (red). Alpha-VI is mode seeking for large $\alpha$ and mass covering for smaller $\alpha$.

**Figure 3:** Sampling variance of the stochastic gradient (averaged over its components) in the optimum, for alpha-divergences (orange, purple, gray), and the proposed PBBVI (green). The variance grows exponentially with the latent dimension $N$ for alpha-VI, and only algebraically for PBBVI.

et al., 2013; Opper, 2015). In variational inference, perturbation theory yields extra terms to the mean-field variational objective which are difficult to calculate analytically. This may be a reason why the methods discussed are not widely adopted by practitioners. In this paper, we emphasize the ease of including perturbative corrections in a black box variational inference framework. Furthermore, in contrast to earlier formulations, our approach yields a strict lower bound to the marginal likelihood which can be conveniently optimized. Our approach is different from the traditional variational perturbation formulation (Kleinert, 2009), which generally does not result in a bound.

## 3 Method

In this section, we present our main contributions. We first present our view of black box variational inference (BBVI) as a form of biased importance sampling in Section 3.1. With this view, we bridge the gap between variational inference and importance sampling. In Section 3.2, we introduce our family of new variational bounds, and analyze their properties further in Section 3.3.

### 3.1 Black Box Variational Inference as Biased Importance Sampling

Consider a probabilistic model with data $\mathbf{x}$, latent variables $\mathbf{z}$, and joint distribution $p(\mathbf{x}, \mathbf{z})$. We are interested in the posterior distribution over the latent variables, $p(\mathbf{z}|\mathbf{x}) = p(\mathbf{x}, \mathbf{z})/p(\mathbf{x})$. This involves the intractable marginal likelihood $p(\mathbf{x})$. In variational inference (Jordan et al., 1999), we instead minimize a divergence measure between a variational distribution $q(\mathbf{z}; \lambda)$ and the posterior. Here, $\lambda$ are parameters of the variational distribution, and we aim to find the parameters $\lambda^*$ that minimize the distance to the posterior. This is equivalent to maximizing a lower bound on the marginal likelihood.

We call the difference between the log variational distribution and the log joint distribution the *interaction energy*,

$$V(\mathbf{z}; \lambda) = \log q(\mathbf{z}; \lambda) - \log p(\mathbf{x}, \mathbf{z}). \tag{2}$$

We use $V$ or $V(\mathbf{z})$ interchangeably to denote $V(\mathbf{z}; \lambda)$, and $q(\mathbf{z})$ to denote $q(\mathbf{z}; \lambda)$, when more convenient. Using this notation, the marginal likelihood is

$$p(\mathbf{x}) = \mathbb{E}_{q(\mathbf{z})}[e^{-V(\mathbf{z})}]. \tag{3}$$

We call $e^{-V(\mathbf{z})} = p(\mathbf{x}, \mathbf{z})/q(\mathbf{z})$ the *importance ratio*, since sampling from $q(\mathbf{z})$ to estimate the right-hand side of Eq. 3 is equivalent to importance sampling. As importance sampling is inefficient in high dimensions, we resort to variational inference. To this end, let $f(\cdot)$ be any concave function defined on the positive reals. We assume furthermore that for all $x > 0$, we have $f(x) \leq x$. Applying Jensen's inequality, we can lower-bound the marginal likelihood,

$$p(\mathbf{x}) \geq f(p(\mathbf{x})) \geq \mathbb{E}_{q(\mathbf{z})}[f(e^{-V(\mathbf{z}; \lambda)})] \equiv \mathcal{L}_f(\lambda). \tag{4}$$

Figure 1 shows exemplary choices of $f$. We maximize $\mathcal{L}_f(\lambda)$ using reparameterization gradients, where the bound is not computed analytically, but rather its gradients are estimated by sampling from

$q(\mathbf{z})$ (Kingma and Welling, 2014). This leads to a stochastic gradient descent scheme, where the noise is a result of the Monte-Carlo estimation of the gradients.

Our approach builds on the insight that black box variational inference is a type of biased importance sampling, where we estimate a lower bound of the marginal likelihood by sampling from a proposal distribution, iteratively improving this distribution. The approach is biased since we do not estimate the exact marginal likelihood but only a lower bound to this quantity. As we argue below, the introduced bias allows us to estimate the bound more easily, because we decrease the variance of this estimator. The choice of the function $f$ thereby trades-off between bias and variance in the following way:

- For $f = id$ being the identity, we obtain *importance sampling*. (See the black line in Figure 1). In this case, Eq. 4 does not depend on the variational parameters, so there is nothing to optimize and we can directly sample from any proposal distribution $q$. Since the expectation under $q$ of the importance ratio $e^{-V(\mathbf{z})}$ gives the exact marginal likelihood, there is no bias. If the model has a large number of latent variables, the importance ratio $e^{-V(\mathbf{z})}$ becomes tightly peaked around the minimum of the interaction energy $V$, resulting in a very high variance of this estimator. Importance sampling is therefore on one extreme end of the bias-variance spectrum.

- For $f = \log$, we obtain the familiar *Kullback-Leibler (KL) bound*. (See the pink line in Figure 1; here we add a constant of 1 for comparison, which does not affect the optimization). Since $f(e^{-V(\mathbf{z})}) = -V(\mathbf{z})$, the bound is

$$\mathcal{L}_{KL}(\lambda) = \mathop{\mathbb{E}}_{q(\mathbf{z})}[-V(\mathbf{z})] = \mathop{\mathbb{E}}_{q(\mathbf{z})}[\log p(\mathbf{x}, \mathbf{z}) - \log q(\mathbf{z})]. \tag{5}$$

  The Monte-Carlo expectation of $\mathbb{E}_q[-V]$ has a much smaller variance than $\mathbb{E}_q[e^{-V}]$, implying efficient learning (Bottou, 2010). However, by replacing $e^{-V}$ with $-V$ we introduce a bias. We can further trade-off less variance for even more bias by dropping the entropy term on the right-hand side of Eq. 5. A flexible enough variational distribution will shrink to zero variance, which completely eliminates the sampling noise. This is equivalent to point-estimation, and is at the opposite end of the bias-variance spectrum.

- Now, consider any $f$ which is between the logarithm and the identity, e.g., the green line in Figure 1 (this is the regularizing function we propose in Section 3.2 below). The more similar $f$ is to the identity, the less biased is our estimate of the marginal likelihood, but the larger the variance. Conversely, the more $f$ behaves like the logarithm, the easier it is to estimate $f(e^{-V(\mathbf{z})})$ by sampling, while at the same time the bias grows.

One example of alternative divergences to the KL divergence that have been discussed in the literature are alpha-divergences (Minka, 2005; Hernandez-Lobato et al., 2016; Li and Turner, 2016; Dieng et al., 2017). Up to a constant, they correspond to the following choice of $f$:

$$f^{(\alpha)}(e^{-V}) \propto e^{-(1-\alpha)V}. \tag{6}$$

The real parameter $\alpha$ determines the distance to the importance sampling case ($\alpha = 0$). As $\alpha$ approaches 1 from below, this bound leads to a better-behaved estimation problem of the Monte-Carlo gradient. However, unless taking the limit of $\alpha \to 1$ (where the objective becomes the KL-bound), $V$ still enters exponentially in the bound. As we show, this leads to a high variance of the gradient estimator in high dimensions (see Figure 3 discussed below). The alpha-divergence bound is therefore similarly as hard to estimate as the marginal likelihood in importance sampling.

Our analysis relies on the observation that expectations of exponentials in $V$ are difficult to estimate, and expectations of polynomials in $V$ are easy to estimate. We derive a family of new variational bounds which are polynomials in $V$, where increasing the order of the polynomial reduces the bias.

### 3.2 Perturbative Black Box Variational Inference

**Perturbative bounds.** We now motivate the family of lower bounds proposed in Eq. 1 in the introduction based on the considerations outlined above. For fixed odd integer $K$ and fixed real value $V_0$, the bound $\mathcal{L}^{(K)}(\lambda, V_0)$ is of the form of Eq. 4 with the following regularizing function $f$:

$$f_{V_0}^{(K)}(x) = e^{-V_0} \sum_{k=0}^{K} \frac{(V_0 + \log x)^k}{k!} \qquad \Longrightarrow \qquad f_{V_0}^{(K)}(e^{-V}) = e^{-V_0} \sum_{k=0}^{K} \frac{(V_0 - V)^k}{k!}. \tag{7}$$

---

**Algorithm 1:** Perturbative Black Box Variational Inference (PBBVI)

---

**Input:** joint probability $p(\mathbf{x}, \mathbf{z})$; order of perturbation $K$ (odd integer); learning rate schedule $\rho_t$; number of Monte Carlo samples $S$; number of training iterations $T$; variational family $q(\mathbf{z}, \lambda)$ that allows for reparameterization gradients, i.e., $\mathbf{z} \sim q(\,\cdot\,, \lambda) \iff \mathbf{z} = g(\boldsymbol{\epsilon}, \lambda)$ where $\boldsymbol{\epsilon} \sim p_\mathrm{n}$ with a fixed noise distribution $p_\mathrm{n}$ and a differentiable reparameterization function $g$.

**Output:** fitted variational parameters $\lambda^*$.

1 initialize $\lambda$ randomly and $V_0 \leftarrow 0$;
2 **for** $t \leftarrow 1$ **to** $T$ **do**
3 $\quad$ draw $S$ samples $\boldsymbol{\epsilon}_1, \ldots, \boldsymbol{\epsilon}_S \sim p_\mathrm{n}$ from the noise distribution;

$\quad$ *// obtain reparameterization gradient estimates using automatic differentiation:*

4 $\quad g_\lambda \leftarrow \hat{\nabla}_\lambda \tilde{\mathcal{L}}^{(K)}(\lambda, V_0) \equiv \nabla_\lambda \left[ \frac{1}{S} \sum_{s=1}^{S} \sum_{k=0}^{K} \frac{1}{k!} \left( \log p(\mathbf{x}, g(\boldsymbol{\epsilon}_s, \lambda)) - \log q(g(\boldsymbol{\epsilon}_s, \lambda); \lambda) + V_0 \right)^k \right]$;

5 $\quad g_{V_0} \leftarrow \hat{\nabla}_{V_0} \tilde{\mathcal{L}}^{(K)}(\lambda, V_0) \equiv \nabla_{V_0} \left[ \frac{1}{S} \sum_{s=1}^{S} \sum_{k=0}^{K} \frac{1}{k!} \left( \log p(\mathbf{x}, g(\boldsymbol{\epsilon}_s, \lambda)) - \log q(g(\boldsymbol{\epsilon}_s, \lambda); \lambda) + V_0 \right)^k \right]$;

$\quad$ *// perform variable updates (see second to last paragraph of Section 3.2):*

6 $\quad \lambda \leftarrow \lambda + \rho_t g_\lambda$;
7 $\quad V_0 \leftarrow V_0 + \rho_t \left[ g_{V_0} - \frac{1}{S} \sum_{s=1}^{S} \sum_{k=0}^{K} \frac{1}{k!} \left( \log p(\mathbf{x}, g(\boldsymbol{\epsilon}_s, \lambda)) - \log q(g(\boldsymbol{\epsilon}_s, \lambda); \lambda) + V_0 \right)^k \right]$;

**end**

---

Here, the second (equivalent) formulation makes it explicit that $f_{V_0}^{(K)}$ is the $K^\text{th}$ order Taylor expansion of its argument $e^{-V}$ in $V$ around some reference energy $V_0$. Figure 1 shows $f_{V_0}^{(K)}(x)$ for $K = 1$ (red) and $K = 3$ (green). The curves are concave and lie below the identity, touching it at $x = e^{-V_0}$. We show in Section 3.3 that these properties extend to every odd $K$ and every $V_0 \in \mathbb{R}$. Therefore, $\mathcal{L}^{(K)}(\lambda, V_0)$ is indeed a lower bound on the marginal likelihood.

The rationale for the design of the regularizing function in Eq. 7 is as follows. On the one hand, the gradients of the resulting bound should be easy to estimate via the reparameterization approach. We achieve low-variance gradient estimates by making $f_{V_0}^{(K)}(e^{-V})$ a polynomial in $V$, i.e., in contrast to the alpha-bound, $V$ never appears in the exponent.

On the other hand, the regularizing function should be close to the identity function so that the resulting bound has low bias. For $K = 1$, we have $\mathcal{L}^{(1)}(\lambda, V_0) = e^{-V_0} \mathbb{E}_q[\log p - \log q + V_0]$. Maximizing $\mathcal{L}^{(1)}$ over $\lambda$ is independent of the value of $V_0$ and equivalent to maximizing the standard KL bound $\mathcal{L}_{KL}$, see Eq. 5, which has low gradient variance and large bias. Increasing the order $K$ to larger odd integers makes the Taylor expansion tighter, leading to a bound with lower bias. In fact, in the limit $K \to \infty$, the right-hand side of Eq. 7 is the series representation of the exponential function, and thus $f_{V_0}^{(K)}$ converges pointwise to the identity. In practice, we propose to set $K$ to a small odd integer larger than 1. Increasing $K$ further reduces the bias, but it comes at the cost of increasing the gradient variance because the random variable $V$ appears in higher orders under the expectation in Eq. 4.

As discussed in Section 3.1, the KL bound $\mathcal{L}_{KL}$ can be derived from a regularizing function $f = \log$ that does not depend on any further parameters like $V_0$. The derivation of the KL bound therefore does not require the first inequality in Eq. 4, and one directly obtains a bound on the model evidence $\log p(\mathbf{x}) \equiv f(p(\mathbf{x}))$ from the second inequality alone. For $K > 1$, the bound $\mathcal{L}^{(K)}(\lambda, V_0)$ depends nontrivially on $V_0$, and we have to employ the first inequality in Eq. 4 in order to make the bounded quantity on the left-hand side independent of $V_0$. This expenses some tightness of the bound but makes the method more flexible by allowing us to optimize over $V_0$ as well, as we describe next.

**Optimization algorithm.** We now propose the perturbative black box variational inference (PBBVI) algorithm. Since $\mathcal{L}^{(K)}(\lambda, V_0)$ is a lower bound on the marginal likelihood for all $\lambda$ and all $V_0$, we can find the values $\lambda^*$ and $V_0^*$ for which the bound is tightest by maximizing simultaneously over $\lambda$ and $V_0$. Algorithm 1 summarizes the PBBVI algorithm. We minimize $-\mathcal{L}^{(K)}(\lambda, V_0)$ using stochastic gradient descent (SGD) with reparameterization gradients and a learning rate $\rho_t$ that decreases with the training iteration $t$ according to Robbins-Monro bounds (Robbins and Monro, 1951). We obtain unbiased gradient estimators (denoted by "$\hat{\nabla}$") using standard techniques: we replace the expectation

$\mathbb{E}_q[\,\cdot\,]$ in Eq. 1 with the empirical average over a fixed number of $S$ samples from $q$, and we calculate the reparameterization gradients with respect to $\lambda$ and $V_0$ using automatic differentiation.

In practice, we typically discard the value of $V_0^*$ once the optimization is converged since we are only interested in the fitted variational parameters $\lambda^*$. However, during the optimization process, $V_0$ is an important auxiliary quantity and the inference algorithm would be inconsistent without an optimization over $V_0$: if we were to extend the model $p(\mathbf{x}, \mathbf{z})$ by an additional observed variable $\tilde{x}$ which is statistically independent of the latent variables $\mathbf{z}$, then the log joint (as a function of $\mathbf{z}$ alone) changes by a constant positive prefactor. The posterior remains unchanged by the constant prefactor, and a consistent VI algorithm must therefore produce the same approximate posterior distribution $q$ for both models. Optimizing over $V_0$ ensures this consistency since the log joint appears in the lower bound only in the combination $\log p(\mathbf{x}, \mathbf{z}) + V_0$. Therefore, a rescaling of the log joint by a constant positive prefactor can be completely absorbed by a change in the reference energy $V_0$.

We observed in our experiments that the reference energy $V_0$ can become very large (in absolute value) for models with many latent variables. To avoid numerical overflow or underflow from the prefactor $e^{-V_0}$, we consider the surrogate objective $\tilde{\mathcal{L}}^{(K)}(\lambda, V_0) \equiv e^{V_0} \mathcal{L}^{(K)}(\lambda, V_0)$. The gradients with respect to $\lambda$ of $\mathcal{L}^{(K)}(\lambda, V_0)$ and $\tilde{\mathcal{L}}^{(K)}(\lambda, V_0)$ are equal up to a positive prefactor, so we can replace the former with the latter in the update step (line 6 in Algorithm 1). The gradient with respect to $V_0$ is $\nabla_{V_0} \mathcal{L}^{(K)}(\lambda, V_0) \propto \nabla_{V_0} \tilde{\mathcal{L}}^{(K)}(\lambda, V_0) - \tilde{\mathcal{L}}^{(K)}(\lambda, V_0)$ (line 7). Using the surrogate $\tilde{\mathcal{L}}^{(K)}(\lambda, V_0)$ avoids numerical underflow or overflow, as well as exponentially increasing or decreasing gradients.

**Mass covering effect.** In Figure 2, we fit a Gaussian distribution to a one-dimensional bimodal target distribution (black line), using different divergences. Compared to BBVI with the standard KL divergence (KLVI, pink line), alpha-divergences are more mode-seeking (purple line) for large values of $\alpha$, and more mass-covering (orange line) for small $\alpha$ (Li and Turner, 2016). Our PBBVI bound ($K = 3$, green line) achieves a similar mass-covering effect as in alpha-divergences, but with associated low-variance reparameterization gradients. This is seen in Figure 3, discussed in Section 4.2, which compares the gradient variances of alpha-VI and PBBVI as a function of dimensions.

### 3.3 Proof of Correctness and Nontriviality of the Bound

To conclude the presentation of the PBBVI algorithm, we prove that the objective in Eq. 1 is indeed a lower bound on the marginal likelihood for all odd orders $K$, and that the bound is nontrivial.

**Correctness.** The lower bound $\mathcal{L}^{(K)}(\lambda, V_0)$ results from inserting the regularizing function $f_{V_0}^{(K)}$ from Eq. 7 into Eq. 4. For odd $K$, it is indeed a valid lower bound because $f_{V_0}^{(K)}$ is concave and lies below the identity. To see this, note that the second derivative $\partial^2 f_{V_0}^{(K)}(x)/\partial x^2 = -e^{-V_0}(V_0 + \log x)^{K-1}/((K-1)!\,x^2)$ is non-positive everywhere for odd $K$. Therefore, the function is concave. Next, consider the function $g(x) = f_{V_0}^{(K)}(x) - x$, which has a stationary point at $x = x_0 \equiv e^{-V_0}$. Since $g$ is also concave, $x_0$ is a global maximum, and thus $g(x) \leq g(x_0) = 0$ for all $x$, implying that $f_{V_0}^{(K)}(x) \leq x$. Thus, for odd $K$, the function $f_{V_0}^{(K)}$ satisfies all requirements for Eq. 4, and $\mathcal{L}^{(K)}(\lambda, V_0) \equiv \mathbb{E}_q[f_{V_0}^{(K)}(e^{-V})]$ is a lower bound on the marginal likelihood. Note that an even order $K$ does not lead to a valid concave regularizing function.

**Nontriviality.** Since the marginal likelihood $p(\mathbf{x})$ is always positive, a lower bound would be trivial if it was negative. We show that once the optimization algorithm has converged, the bound at the optimum is always positive. At the optimum, all gradients vanish. By setting the derivative with respect to $V_0$ of the right-hand side of Eq. 1 to zero we find that $\mathbb{E}_{q^*}[(V_0^* - V)^K] = 0$, where $q^* \equiv q(\,\cdot\,; \lambda^*)$ is the variational distribution at the optimum. Thus, the lower bound at the optimum is $\mathcal{L}(\lambda^*, V_0^*) = e^{-V_0} \mathbb{E}_{q^*}[h(V)]$ with $h(V) = \sum_{k=0}^{K-1} \frac{1}{k!}(V_0^* - V)^k$, where the sum runs only to $K - 1$ because the term with $k = K$ vanishes at $V_0 = V_0^*$. We show that $h(V)$ is positive for all $V$. If $K = 1$, then $h(V) = 1$ is a positive constant. For $K \geq 3$, $h(V)$ is a polynomial in $V$ of even order $K - 1$, whose highest order term has a positive coefficient $1/(K-1)!$. Therefore, as $V \to \pm\infty$, the function $h(V)$ goes to positive infinity and it thus has a global minimum at some value $\tilde{V} \in \mathbb{R}$. At the global minimum, its derivative vanishes, $0 = \nabla_{\tilde{V}} h(\tilde{V}) = -\sum_{k=0}^{K-2} \frac{1}{k!}(V_0^* - \tilde{V})^k$. Thus, at the global minimum of the polynomial $h$, all terms except the highest order term cancel, and we find $h(\tilde{V}) = \frac{1}{(K-1)!}(V_0^* - \tilde{V})^{K-1} \geq 0$, which is nonnegative because $K - 1$ is even. The case $h(\tilde{V}) = 0$ is achieved if and only if $\tilde{V} = V_0^*$, but this would violate the condition $\nabla_{\tilde{V}} h(\tilde{V}) = 0$. Therefore,

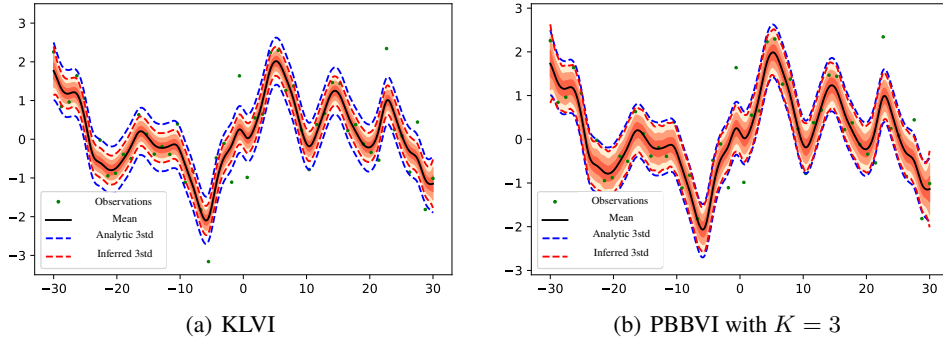

| (a) KLVI | (b) PBBVI with $K = 3$ |

**Figure 4:** Gaussian process regression on synthetic data (green dots). Three standard deviations are shown in varying shades of orange. The blue dashed lines show three standard deviations of the true posterior. The red dashed lines show the inferred three standard deviations using KLVI (a) and PBBVI (b). We see that the results from our proposed PBBVI are close to the analytic solution while traditional KLVI underestimates the variances.

| Method | Avg variances |
|---|---|
| Analytic | 0.0415 |
| KLVI | 0.0176 |
| PBBVI | 0.0355 |

| Data set | Crab | Pima | Heart | Sonar |
|---|---|---|---|---|
| KLVI | 0.22 | 0.245 | 0.148 | 0.212 |
| PBBVI | **0.11** | **0.240** | **0.1333** | **0.1731** |

**Table 1:** Average variances across training examples in the synthetic data experiment. The closer to the analytic solution, the better.

**Table 2:** Error rate of GP classification on the test set. The lower the better. Our proposed PBBVI consistently obtains better classification results.

$h(\tilde{V})$ is strictly positive, and since $\tilde{V}$ is a global minimum of $h$, we have $h(V) \geq h(\tilde{V}) > 0$ for all $V \in \mathbb{R}$. Inserting into the expression for $\mathcal{L}(\lambda^*, V_0^*)$ concludes the proof that the lower bound at the optimum is positive.

## 4 Experiments

We evaluate PBBVI with different models. First we investigate its behavior in a controlled setup of Gaussian processes on synthetic data (Section 4.1). We then evaluate PBBVI based on a classification task using Gaussian processes classifiers, where we use data from the UCI machine learning repository (Section 4.2). This is a Bayesian non-conjugate setup where black box inference is required. Finally, we use an experiment with the variational autoencoder (VAE) to explore our approach on a deep generative model (Section 4.3). This experiment is carried out on MNIST data. We use the perturbative order $K = 3$ for all experiments with PBBVI. This corresponds to the lowest order beyond standard KLVI, since KLVI is equivalent to PBBVI with $K = 1$, and $K$ has to be an odd integer. Across all the experiments, PBBVI demonstrates advantages based on different metrics.

### 4.1 GP Regression on Synthetic Data

In this section, we inspect the inference behavior using a synthetic data set with Gaussian processes (GP). We generate the data according to a Gaussian noise distribution centered around a mixture of sinusoids, and sample 50 data points (green dots in Figure 4). We then use a GP to model the data, thus assuming the generative process $f \sim \mathcal{GP}(0, \Lambda)$ and $y_i \sim \mathcal{N}(f_i, \epsilon)$.

We first compute an analytic solution of the posterior of the GP, (three standard deviations shown in blue dashed lines) and compare it to approximate posteriors obtained by KLVI (Figure 4 (a)) and the proposed PBBVI (Figure 4 (b)). The results from PBBVI are almost identical to the analytic solution. In contrast, KLVI underestimates the posterior variance. This is consistent with Table 1, which shows the average diagonal variances. PBBVI results are much closer to the exact posterior variances.

### 4.2 Gaussian Process Classification

We evaluate the performance of PBBVI and KLVI on a GP classification task. Since the model is non-conjugate, no analytical baseline is available in this case. We model the data with the following generative process:

$$f \sim \mathcal{GP}(0, \Lambda), \quad z_i = \sigma(f_i), \quad y_i \sim Bern(z_i).$$

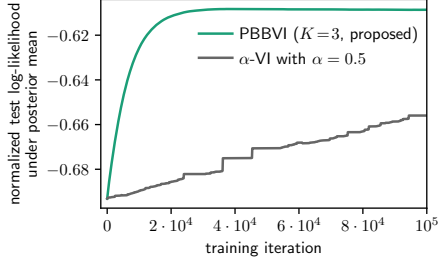
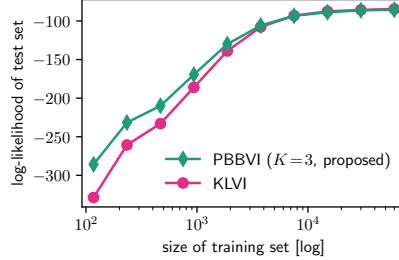

**Figure 5:** Test log-likelihood (normalized by the number of test points) as a function of training iterations using GP classification on the Sonar data set. PBBVI converges faster than alpha-VI even though we tuned the number of Monte Carlo samples per training step (100) and the constant learning rate ($10^{-5}$) so as to maximize the performance of alpha-VI on a validation set.

**Figure 6:** Predictive likelihood of a VAE trained on different sizes of the data. The training data are randomly sampled subsets of the MNIST training set. The higher value the better. Our proposed PBBVI method outperforms KLVI mainly when the size of the training data set is small. The fewer the training data, the more advantage PBBVI obtains.

Above, $\Lambda$ is the GP kernel, $\sigma$ indicates the sigmoid function, and $Bern$ indicates the Bernoulli distribution. We furthermore use the Matern 32 kernel,

$$\Lambda_{ij} = s^2 (1 + \frac{\sqrt{3}\, r_{ij}}{l}) \exp(-\frac{\sqrt{3}\, r_{ij}}{l}), \quad r_{ij} = \sqrt{(x_i - x_j)^T (x_i - x_j)}.$$

**Data.** We use four data sets from the UCI machine learning repository, suitable for binary classification: Crab (200 datapoints), Pima (768 datapoints), Heart (270 datapoints), and Sonar (208 datapoints). We randomly split each of the data sets into two halves. One half is used for training and the other half is used for testing. We set the hyper parameters $s = 1$ and $l = \sqrt{D}/2$ throughout all experiments, where $D$ is the dimensionality of input $x$.

Table 2 shows the classification performance (error rate) for these data sets. Our proposed PBBVI consistently performs better than the traditional KLVI.

**Convergence speed comparison.** We also carry out a comparison in terms of speed of convergence, focusing on PBBVI and alpha-divergence VI. Our results indicate that the smaller variance of the reparameterization gradient leads to faster convergence of the optimization algorithm.

We train the GP classifier from Section 4.2 on the Sonar UCI data set using a constant learning rate. Figure 5 shows the test log-likelihood under the posterior mean as a function of training iterations. We split the data set into equally sized training, validation, and test sets. We then tune the learning rate and the number of Monte Carlo samples per gradient step to obtain optimal performance on the validation set after minimizing the alpha-divergence with a fixed budget of random samples. We use $\alpha = 0.5$ here; smaller values of $\alpha$ lead to even slower convergence. We optimize the PBBVI lower bound using the same learning rate and number of Monte Carlo samples. The final test error rate is 22% on an approximately balanced data set. PBBVI converges an order of magnitude faster.

Figure 3 in Section 3 provides more insight in the scaling of the gradient variance. Here, we fit GP regression models on synthetically generated data by maximizing the PBBVI lower bound and the alpha-VI lower bound with $\alpha \in \{0.2, 0.5, 2\}$. We generate a separate synthetic data set for each $N \in \{1, \ldots, 200\}$ by drawing $N$ random data points around a sinusoidal curve. For each $N$, we fit a one-dimensional GP regression with PBBVI and alpha-VI, respectively, using the same data set for both methods. The variational distribution is a fully factorized Gaussian with $N$ latent variables. After convergence, we estimate the sampling variance of the gradient of each lower bound with respect to the posterior mean. We calculate the empirical variance of the gradient based on $10^5$ samples from $q$, and we average over the $N$ coordinates. Figure 3 shows the average sampling variance as a function of $N$ on a logarithmic scale. The variance of the gradient of the alpha-VI bound grows exponentially in the number of latent variables. By contrast, we find only algebraic growth for PBBVI.

## 4.3 Variational Autoencoder

We experiment on Variational Autoencoders (VAEs), and we compare the PBBVI and the KLVI bound in terms of predictive likelihoods on held-out data (Kingma and Welling, 2014). Autoencoders compress unlabeled training data into low-dimensional representations by fitting it to an encoder-decoder model that maps the data to itself. These models are prone to learning the identity function

when the hyperparameters are not carefully tuned, or when the network is too expressive, especially for a moderately sized training set. VAEs are designed to partially avoid this problem by estimating the uncertainty that is associated with each data point in the latent space. It is therefore important that the inference method does not underestimate posterior variances. We show that, for small data sets, training a VAE by maximizing the PBBVI lower bound leads to higher predictive likelihoods than maximizing the KLVI lower bound.

We train the VAE on the MNIST data set of handwritten digits (LeCun et al., 1998). We build on the publicly available implementation by Burda et al. (2016) and also use the same architecture and hyperparamters, with $L = 2$ stochastic layers and $S = 5$ samples from the variational distribution per gradient step. The model has 100 latent units in the first stochastic layer and 50 latent units in the second stochastic layer.

The VAE model factorizes over all data points. We train it by stochastically maximizing the sum of the PBBVI lower bounds for all data points using a minibatch size of 20. The VAE amortizes the gradient signal across data points by training inference networks. The inference networks express the mean and variance of the variational distribution as a function of the data point. We add an additional inference network that learns the mapping from a data point to the reference energy $V_0$. Here, we use a network with four fully connected hidden layers of 200, 200, 100, and 50 units, respectively.

MNIST contains 60,000 training images. To test our approach on smaller-scale data where Bayesian uncertainty matters more, we evaluate the test likelihood after training the model on randomly sampled fractions of the training set. We use the same training schedules as in the publicly available implementation, keeping the total number of training iterations independent of the size of the training set. Different to the original implementation, we shuffle the training set before each training epoch as this turns out to increase the performance for both our method and the baseline.

Figure 6 shows the predictive log-likelihood of the whole test set, where the VAE is trained on random subsets of different sizes of the training set. We use the same subset to train with PBBVI and KLVI for each training set size. PBBVI leads to a higher predictive likelihood than traditional KLVI on subsets of the data. We explain this finding with our observation that the variational distributions obtained from PBBVI capture more of the posterior variance. As the size of the training set grows—and the posterior uncertainty decreases—the performance of KLVI catches up with PBBVI.

As a potential explanation why PBBVI converges to the KLVI result for large training sets, we note that $\mathbb{E}_{q^*}[(V_0^* - V)^3] = 0$ at the optimal variational distribution $q^*$ and reference energy $V_0^*$ (see Section 3.3). If $V$ becomes a symmetric random variable (such as a Gaussian) in the limit of a large training set, then this implies that $\mathbb{E}_{q^*}[V] = V_0^*$, and PBBVI reduces to KLVI for large training sets.

## 5 Conclusion

We first presented a view on black box variational inference as a form of biased importance sampling, where we can trade-off bias versus variance by the choice of divergence. Bias refers to the deviation of the bound from the true marginal likelihood, and variance refers to its reparameterization gradient estimator. We then proposed a family of new variational bounds that connect to variational perturbation theory, and which include corrections to the standard Kullback-Leibler bound. Our proposed PBBVI bound converges to the true marginal likelihood for large order $K$ of the perturbative expansion, and we showed both theoretically and experimentally that it has lower-variance reparameterization gradients compared to alpha-VI. In order to scale up our method to massive data sets, future work will explore stochastic versions of PBBVI. Since the PBBVI bound contains interaction terms between all data points, breaking it up into mini-batches is non-straightforward. Besides, while our experiments used a fixed perturbative order of $K = 3$, it could be beneficial to increase the perturbative order at some point during the training cycle once an empirical estimate of the gradient variance drops below a certain threshold. Furthermore, the PBBVI and alpha-bounds can also be combined, such that PBBVI further approximates alpha-VI. This could lead to promising results on large data sets where traditional alpha-VI is hard to optimize due to its variance, and traditional PBBVI converges to KLVI. As a final remark, a tighter variational bound is not guaranteed to always result in a better posterior approximation since the variational family limits the quality of the solution. However, in the context of variational EM, where one performs gradient-based hyperparameter optimization on the log marginal likelihood, our bound gives more reliable results since higher orders of $K$ can be assumed to approximate the marginal likelihood better.

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
