[Supplementary Material]

# Supplementary Material to "Perturbative Black Box Variational Inference"

**Robert Bamler**[*]
Disney Research
Pittsburgh, USA

**Cheng Zhang**[*]
Disney Research
Pittsburgh, USA

**Manfred Opper**
TU Berlin
Berlin, Germany

**Stephan Mandt**[*]
Disney Research
Pittsburgh, USA

firstname.lastname@{disneyresearch.com, tu-berlin.de}

## Proof that PBBVI minimizes a divergence

In this supplement we show that perturbative black box variational inference (PBBVI) minimizes a valid divergence from the variational distribution $q(\mathbf{z})$ to the true posterior distribution $p(\mathbf{z}|\mathbf{x})$. This has the important consequence that PBBVI converges to exact inference in the limit of an arbitrarily flexible variational family. In contrast to traditional Kullback-Leibler variational inference (KLVI) and $\alpha$-VI, the divergence minimized by PBBVI depends on the choice of variational family.

Let $f$ be any regularizing function that satisfies the conditions for Eq. 4 of the main text (i.e., $f$ is concave and smaller than the identity). Let $\mathcal{L}_f(q)$ be the associated lower bound on the marginal likelihood. We prefer this notation over the notation $\mathcal{L}_f(\lambda)$ used in the main text here, because, at this stage, we do not restrict $q$ to a specific family of variational distributions indexed by $\lambda$. We define a divergence $D_f$ from $q$ to the true posterior $p(\mathbf{z}|\mathbf{x})$ by

$$D_f(p||q) \equiv f(p(\mathbf{x})) - \mathcal{L}_f(q). \tag{S1}$$

Here, the model marginal likelihood $p(\mathbf{x})$ is unknown to us, but it is a well-defined constant assuming that the model parameters are kept constant. We show that $D_f$ is indeed a valid divergence. From Eq. 4 of the main text, we find $\mathcal{L}_f(q) \leq f(p(\mathbf{x}))$ and therefore $D_f$ is non-negative. The lower bound $\mathcal{L}_f(q)$ is defined in Eqs. 2 and 4 of our paper as

$$\mathcal{L}_f(q) \equiv \mathbb{E}_{q(\mathbf{z})}\left[ f\left( \frac{p(\mathbf{x}, \mathbf{z})}{q(\mathbf{z})} \right) \right]. \tag{S2}$$

Setting $q(\mathbf{z})$ to the true posterior, $p(\mathbf{x}, \mathbf{z})/p(\mathbf{x})$, yields $\mathcal{L}_f(q) = f(p(\mathbf{x}))$, and therefore sets $D_f(p||q)$ to zero. Thus, $D_f$ is indeed a valid divergence.

Consider now the specific family of regularizing functions $f_{V_0}^{(K)}$ defined in Eq. 7 of the main text. We now also restrict $q$ to be a member of some predefined variational family parameterized by $\lambda$. Maximizing the corresponding lower bound $\mathcal{L}^{(K)}(\lambda, V_0) \equiv \mathcal{L}_{f_{V_0}^{(K)}}(q)$ simultaneously over $\lambda$ and $V_0$ yields an optimal reference energy $V_0^*$ and an optimal member $q^* \equiv q(\cdot, \lambda^*)$ of the variational family. Both depend not only on the model but also on the variational family to which $q$ is restricted. Evidently, $q^*$ is the member of the variational family that minimizes the divergence

$$D_{f_{V_0^*}^{(K)}}(p||q) = f_{V_0^*}^{(K)}(p(\mathbf{x})) - \mathcal{L}_{f_{V_0^*}^{(K)}}(q). \tag{S3}$$

Here, the first term on the right-hand side is a constant (since $V_0^*$ is). Its value is not known to us, but well defined. Thus, PBBVI minimizes a valid divergence to the true posterior. As a practical consequence this implies that the exact maximum of the PBBVI lower bound is the true posterior if the variational family is sufficiently flexible to contain it. Note that the choice of divergence that PBBVI minimizes depends on the perturbative order $K$, and also on the model and the variational family (via their influence on $V_0^*$).

---

[*]Equal contribution