[Reviews · NeurIPS 2017]

Reviewer 1



The authors present a novel variation of the black-box variational inference algorithm that uses a non-standard optimization target derived using perturbation theory. The paper is well-written and carefully explains its relationship to existing work in this area. The exposition is easy to follow and the technical results are sound. The experiments are sensible and compare the method described by the authors to current state-of-the-art, in terms of either accuracy or speed as appropriate, in typical settings where variational inference would be employed. The paper should be interesting broadly to users of variational inference. I found the authors' perspective on variational inference as a form of importance sampling to be illuminating, although there is something that bothers me about it. If we view variational inference as a method to approximate the marginal likelihood the whole story about bias-variance trade-off makes perfect sense. However, if we view variational inference as a method to approximate the posterior distribution, things are much less clear to me. In particular, f = id corresponds to an unbiased estimator so it should be the one that performs best in the limit of infinite computation where the expectation with respect to q can be evaluated exactly. But in that case the lower bound is exact regardless of the proposal/variational distribution chosen so the gradient with respect to variational parameters is always zero and we can't perform any optimization. Am I missing something here?

Reviewer 2



Summary: The authors present a new variational objective for approximate Bayesian inference. The variational objective is nicely framed as an interpolation between classic importance sampling and the traditional ELBO-based variational inference. Properties of the variance of importance sampling estimator and ELBO estimators are studied and leveraged to create a better marginal likelihood bound with tractable variance properties. The new bound is based on a low-degree polynomial of the log-importance weight (termed the interaction energy). The traditional ELBO estimator is expressed as a first order polynomial in their more general framework. The authors then test out the idea on a Gaussian process regression problem and a Variational autoencoder. Quality: I enjoyed this paper --- I thought the idea was original, the presentation is clear, and the experiments were convincing. The paper appears to be technically correct, and the method itself appears to be effective. Clarity: This paper was a pleasure to read. Not only was this paper extremely clear and well written, the authors very nicely frame their work in the context of other current and previous research. Originality: While establishing new lower bounds on the marginal likelihood is a common subject at this point, the authors manage to approach this with originality. Impact: I think this paper has the potential to be highly impactful --- the spectrum drawn from importance sampling to KLVI is an effective way of framing these ideas for future research. Questions/Comments: - Figure 1: should the legend text "KLVI: f(x) = 1 + log(x)" read "f(x) = log(x)" ? I believe that would cohere with the bullet point on line 124. - How does the variance of the marginal likelihood bound estimators relate to the variance of the gradients of those estimators wrt variational params? KLVI reparameterization gradients can have some unintuitive irreducibility (Roeder et al, https://arxiv.org/abs/1703.09194); is this the case for PVI?

Reviewer 3



This work casts black box variational inference as a biased importance sampling estimator of the marginal likelihood. In this view, the role of the optimal variational posterior is to minimize the bias. The central contribution is a bound defined by a third order Taylor expansion of the importance weight. The experiments explore the properties of this bound in inference in GPs and maximum likelihood estimation in VAEs. The paper is well written overall and easy to understand. Still, the contribution is relatively minor and doesn't really provide large empirical wins in large data settings. Specific questions/comments for the authors: (1) In what sense is KLVI at the low variance large bias end of the bias/var spectrum? You can clearly have bounds that have 0 variance and infinite bias. More importantly, are there sensible lower variance and higher bias bounds than KLVI? (2) tilde L on line 173 is not apparently defined anywhere. (3) Can you give any characterization of the optimal q as it relates to the true posterior? That is one of the nice features of the ELBO. (4) It seems like the convergence comparison of PVI and alpha-VI could use a few more points of comparison. Why was alpha=2 chosen? how does it compare to other choices of alpha? Is it not possible to use smaller positive alphas? (5) Can the authors comment a bit more on the convergence of PVI to KLVI in the large data limit? I don't fully follow the argument, and seems to be a relatively negative result for the context in which this may be used.